# Quantum-chemical insights from deep tensor neural networks

Kristof T. Schütt[1], Farhad Arbabzadah[1], Stefan Chmiela[1], Klaus R. Müller[1,2] & Alexandre Tkatchenko[3,4]

Learning from data has led to paradigm shifts in a multitude of disciplines, including web, text and image search, speech recognition, as well as bioinformatics. Can machine learning enable similar breakthroughs in understanding quantum many-body systems? Here we develop an efficient deep learning approach that enables spatially and chemically resolved insights into quantum-mechanical observables of molecular systems. We unify concepts from many-body Hamiltonians with purpose-designed deep tensor neural networks, which leads to size-extensive and uniformly accurate ($1\,\text{kcal}\,\text{mol}^{-1}$) predictions in compositional and configurational chemical space for molecules of intermediate size. As an example of chemical relevance, the model reveals a classification of aromatic rings with respect to their stability. Further applications of our model for predicting atomic energies and local chemical potentials in molecules, reliable isomer energies, and molecules with peculiar electronic structure demonstrate the potential of machine learning for revealing insights into complex quantum-chemical systems.

[1] Machine Learning Group, Technische Universität Berlin, Marchstr. 23, 10587 Berlin, Germany. [2] Department of Brain and Cognitive Engineering, Korea University, Anam-dong, Seongbuk-gu, Seoul 136-713, Republic of Korea. [3] Theory Department, Fritz-Haber-Institut der Max-Planck-Gesellschaft, Faradayweg 4-6, D-14195 Berlin, Germany. [4] Physics and Materials Science Research Unit, University of Luxembourg, Luxembourg,, L-1511 Luxembourg. Correspondence and requests for materials should be addressed to K.R.M. (email: klaus-robert.mueller@tu-berlin.de) or to A.T. (email: alexandre.tkatchenko@uni.lu).

Chemistry permeates all aspects of our life, from the development of new drugs to the food that we consume and materials we use on a daily basis. Chemists rely on empirical observations based on creative and painstaking experimentation that leads to eventual discoveries of molecules and materials with desired properties and mechanisms to synthesize them. Many discoveries in chemistry can be guided by searching large databases of experimental or computational molecular structures and properties by using concepts based on chemical similarity. Because the structure and properties of molecules are determined by the laws of quantum mechanics, ultimately chemical discovery must be based on fundamental quantum principles. Indeed, electronic structure calculations and intelligent data analysis (machine learning) have recently been combined aiming towards the goal of accelerated discovery of chemicals with desired properties[1–8]. However, so far the majority of these pioneering efforts have focused on the construction of reduced models trained on large data sets of density-functional theory calculations.

In this work, we develop an efficient deep learning approach that enables spatially and chemically resolved insights into quantum-mechanical properties of molecular systems beyond those trivially contained in the training dataset. Obviously, computational models are not predictive if they lack accuracy. In addition to being interpretable, size-extensive and efficient, our deep tensor neural network (DTNN) approach is uniformly accurate ($1 \, \text{kcal} \, \text{mol}^{-1}$) throughout compositional and configurational chemical space. On the more fundamental side, the mathematical construction of the DTNN model provides statistically rigorous partitioning of extensive molecular properties into atomic contributions—a long-standing challenge for quantum-mechanical calculations of molecules.

## Results

**Molecular deep tensor neural networks.** It is common to use a carefully chosen representation of the problem at hand as a basis for machine learning[9–11]. For example, molecules can be represented as Coulomb matrices[7,12,13], scattering transforms[14], bags of bonds[15], smooth overlap of atomic positions[16,17] or generalized symmetry functions[18,19]. Kernel-based learning of molecular properties transforms these representations non-linearly by virtue of kernel functions. In contrast, deep neural networks[20] are able to infer the underlying regularities and learn an efficient representation in a layer-wise fashion[21].

Molecular properties are governed by the laws of quantum mechanics, which yield the remarkable flexibility of chemical systems, but also impose constraints on the behaviour of bonding in molecules. The approach presented here utilizes the many-body Hamiltonian concept for the construction of the DTNN architecture (Fig. 1), embracing the principles of quantum chemistry, while maintaining the full flexibility of a complex data-driven learning machine.

DTNN receives molecular structures through a vector of nuclear charges $\mathbf{Z}$ and a matrix of atomic distances $D$ ensuring rotational and translational invariance by construction (Fig. 1a). The distances are expanded in a Gaussian basis, yielding a feature vector $\hat{\mathbf{d}}_{ij} \in \mathbb{R}^G$, which accounts for the different nature of interactions at various distance regimes. Similar approaches have been applied to the entries of the Coulomb matrix for the prediction of molecular properties before[12].

The total energy $E_M$ for the molecule $M$ composed of $N$ atoms is written as a sum over $N$ atomic energy contributions $E_i$, thus satisfying permutational invariance with respect to atom indexing. Each atom $i$ is represented by a coefficient vector $\mathbf{c} \in \mathbb{R}^B$, where $B$ is the number of basis functions, or features. Motivated

by quantum-chemical atomic basis set expansions, we assign an atom type-specific descriptor vector $\mathbf{c}_{Z_i}$ to these coefficients $\mathbf{c}_i^{(0)}$. Subsequently, this atomic expansion is repeatedly refined by pairwise interactions with the surrounding atoms

$$\mathbf{c}_i^{(t+1)} = \mathbf{c}_i^{(t)} + \sum_{j \neq i} \mathbf{v}_{ij}, \qquad (1)$$

where the interaction term $\mathbf{v}_{ij}$ reflects the influence of atom $j$ at a distance $D_{ij}$ on atom $i$. Note that this refinement step is seamlessly integrated into the architecture of the molecular DTNN, and is therefore adapted throughout the learning process. In Supplementary Discussion, we show the relation to convolutional neural networks that have been applied to images, speech and text with great success because of their ability to capture local structure[22–27]. Considering a molecule as a graph, $T$ refinements of the coefficient vectors are comprised of all walks of length $T$ through the molecule ending at the corresponding atom[28,29]. From the point of view of many-body interatomic interactions, subsequent refinement steps $t$ correlate atomic neighbourhoods with increasing complexity.

While the initial atomic representations only consider isolated atoms, the interaction terms characterize how the basis functions of two atoms overlap with each other at a certain distance. Each refinement step is supposed to reduce these overlaps, thereby embedding the atoms of the molecule into their chemical environment. Following this procedure, the DTNN implicitly learns an atom-centered basis that is unique and efficient with respect to the property to be predicted.

Non-linear coupling between the atomic vector features and the interatomic distances is achieved by a tensor layer[30–32], such that the coefficient $k$ of the refinement is given by

$$v_{ijk} = \tanh\!\left( \mathbf{c}_j^{(t)} V_k \hat{\mathbf{d}}_{ij} + \left( W^{\mathrm{c}} \mathbf{c}_j^{(t)} \right)_k + \left( W^{\mathrm{d}} \hat{\mathbf{d}}_{ij} \right)_k + b_k \right), \qquad (2)$$

where $b_k$ is the bias of feature $k$ and $W^{\mathrm{c}}$ and $W^{\mathrm{d}}$ are the weights of atom representation and distance, respectively. The slice $V_k$ of the parameter tensor $V \in \mathbb{R}^{B \times B \times G}$ combines the inputs multiplicatively. Since $V$ incorporates many parameters, using this kind of layer is both computationally expensive as well as prone to overfitting. Therefore, we employ a low-rank tensor factorization, as described in (ref. 33), such that

$$\mathbf{v}_{ij} = \tanh\!\left[ W^{\mathrm{fc}} \left( \left( W^{\mathrm{cf}} \mathbf{c}_j + \mathbf{b}^{\mathrm{f}_1} \right) \circ \left( W^{\mathrm{df}} \hat{\mathbf{d}}_{ij} + \mathbf{b}^{\mathrm{f}_2} \right) \right) \right], \qquad (3)$$

where '$\circ$' represents element-wise multiplication, while $W^{\mathrm{cf}}$, $\mathbf{b}^{\mathrm{f}_1}$, $W^{\mathrm{df}}$, $\mathbf{b}^{\mathrm{f}_2}$ and $W^{\mathrm{fc}}$ are the weight matrices and corresponding biases of atom representations, distances and resulting factors, respectively. As the dimensionality of $W^{\mathrm{cf}} \mathbf{c}_j$ and $W^{\mathrm{df}} \hat{\mathbf{d}}_{ij}$ corresponds to the number of factors, choosing only a few drastically decreases the number of parameters, thus solving both issues of the tensor layer at once.

Arriving at the final embedding after a given number of interaction refinements, two fully-connected layers predict an energy contribution from each atomic coefficient vector, such that their sum corresponds to the total molecular energy $E_M$. Therefore, the DTNN architecture scales with the number of atoms in a molecule, fully capturing the extensive nature of the energy. All weights, biases, as well as the atom type-specific descriptors were initialized randomly and trained using stochastic gradient descent.

**Learning molecular energies.** To demonstrate the versatility of the proposed DTNN, we train models with up to three interaction passes $T = 3$ for both compositional and configurational degrees of freedom in molecular systems. The DTNN accuracy saturates at $T = 3$, and leads to a strong correlation between atoms in

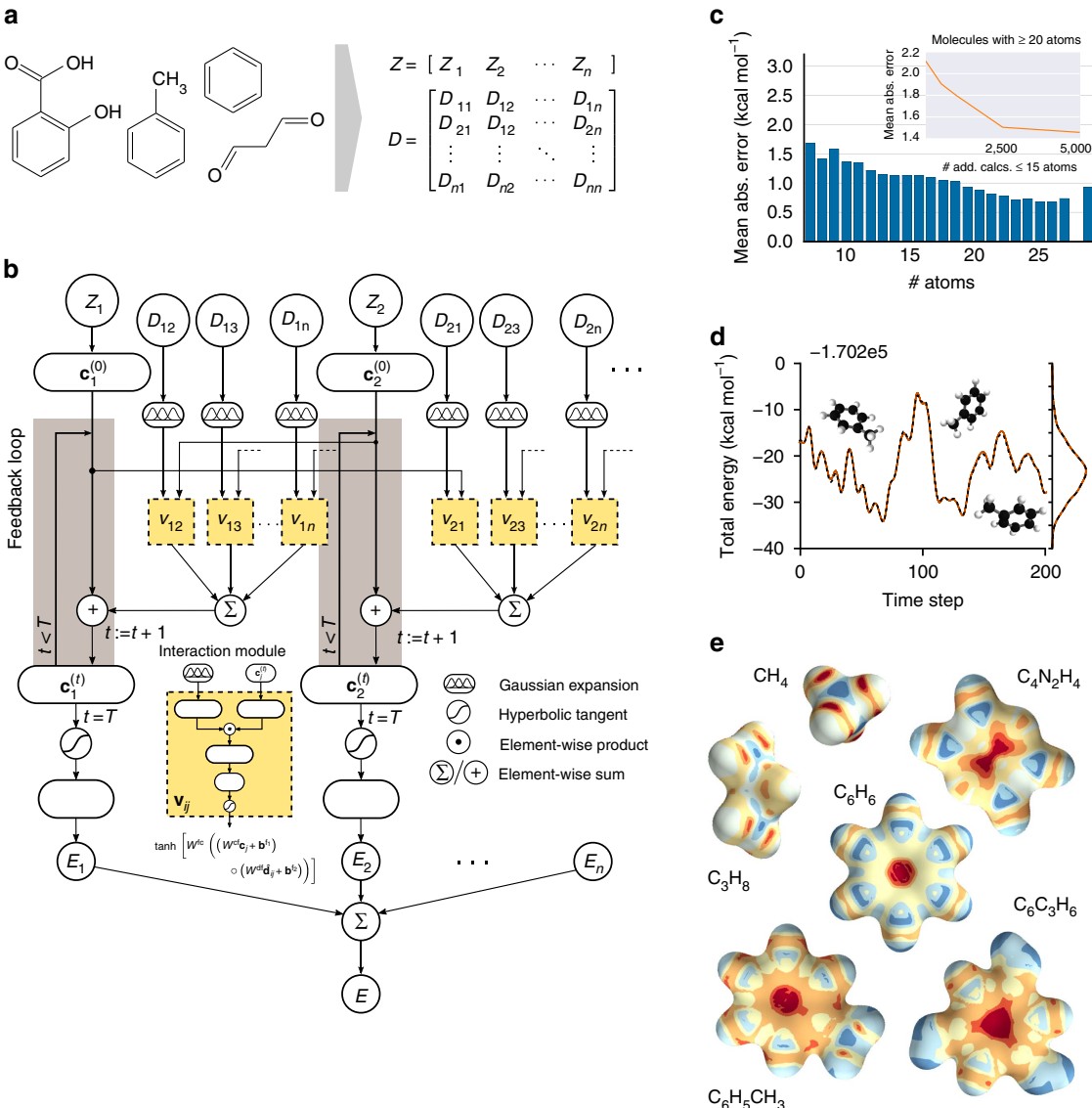

**Figure 1 | Prediction and explanation of molecular energies with a deep tensor neural network. (a)** Molecules are encoded as input for the neural network by a vector of nuclear charges and an inter-atomic distance matrix. This description is complete and invariant to rotation and translation.
**(b)** Illustration of the network architecture. Each atom type corresponds to a vector of coefficients $c_i^{(0)}$, which is repeatedly refined by interactions $v_{ij}$. The interactions depend on the current representation $c_j^{(t)}$, as well as the distance $D_{ij}$ to an atom $j$. After $T$ iterations, an energy contribution $E_i$ is predicted for the final coefficient vector $c_i^{(T)}$. The molecular energy $E$ is the sum over these atomic contributions. **(c)** Mean absolute errors of predictions for the GDB-9 dataset of 133,885 molecules as a function of the number of atoms. The employed neural network uses two interaction passes ($T = 2$) and 50,000 reference calculation during training. The inset shows the error of an equivalent network trained on 5,000 GDB-9 molecules with 20 or more atoms, as small molecules with 15 or less atoms are added to the training set. **(d)** Extract from the calculated (black) and predicted (orange) molecular dynamics trajectory of toluene. The curve on the right shows the agreement of the predicted and calculated energy distributions. **(e)** Energy contribution $E_{probe}$ (or local chemical potential $\Omega_H^M(\mathbf{r})$, see text) of a hydrogen test charge on a $\sum_i \|\mathbf{r} - \mathbf{r}_i\|^{-2}$ isosurface for various molecules from the GDB-9 dataset for a DTNN model with $T = 2$.

molecules, as can be visualized by the complexity of the potential learned by the network (Fig. 1e). For training, we employ chemically diverse data sets of equilibrium molecular structures, as well as molecular dynamics (MD) trajectories for small molecules. We employ two subsets of the GDB-13 database[34,35] referred to as GDB-7, including >7,000 molecules with up to seven heavy (C, N, O, F) atoms, and GDB-9, consisting of 133,885 molecules with up to nine heavy atoms[36]. In both cases, the learning task is to predict the molecular total energy calculated with density-functional theory (DFT). All GDB molecules are stable and synthetically accessible according to organic chemistry rules[35]. Molecular features such as functional groups or

signatures include single, double and triple bonds; (hetero-) cycles, carboxy, cyanide, amide, amine, alcohol, epoxy, sulphide, ether, ester, chloride, aliphatic and aromatic groups. For each of the many possible stoichiometries, many constitutional isomers are considered, each being represented only by a low-energy conformational isomer.

As Supplementary Table 1 demonstrates, DTNN achieves a mean absolute error of 1.0 kcal mol$^{-1}$ on both GDB data sets, training on 5.8 k GDB-7 (80%) and 25 k (20%) GDB-9 reference calculations, respectively. Figure 1c shows the performance on GDB-9 depending on the size of the molecule. We observe that larger molecules have lower errors because of their

abundance in the training data. However, when predicting larger molecules than present in the training set, the errors increase. This is because the molecules in the GDB-9 set are quite small, so we considered all atoms to be in each other's chemical environment. Imposing a distance cutoff to interatomic interactions of 3 Å leads to a 0.1 kcal mol$^{-1}$ increase in the error. However, this distance cutoff restricts only the direct interactions considered in the refinement steps. With multiple refinements, the effective cutoff increases by a factor of $T$ because of indirect interactions over multiple atoms. Given large enough molecules, so that a reasonable distance cutoff can be chosen, scaling to larger molecules will require only to have well-represented local environments. For now, we observe that at least a few larger molecules are needed to achieve a good prediction accuracy. Following this train of thought, we trained the network on a restricted subset of 5 k molecules with > 20 atoms. By adding smaller molecules to the training set, we are able to reduce the test error from 2.1 kcal mol$^{-1}$ to < 1.5 kcal mol$^{-1}$ (see inset in Fig. 1c). This result demonstrates that our model is able to transfer knowledge learned from small molecules to larger molecules with diverse functional groups.

While only encompassing conformations of a single molecule, reproducing MD simulation trajectories poses a radically different challenge to predicting energies of purely equilibrium structures. We learned potential energies for MD trajectories of benzene, toluene, malonaldehyde and salicylic acid, carried out at a rather high temperature of 500 K to achieve exhaustive exploration of the potential-energy surface of such small molecules. The neural network yields mean absolute errors of 0.05, 0.18, 0.17 and 0.39 kcal mol$^{-1}$ for these molecules, respectively (Supplementary Table 1). Figure 1d shows the excellent agreement between the DFT and DTNN MD trajectory of toluene, as well as the corresponding energy distributions. The DTNN errors are much smaller than the energy of thermal fluctuations at room temperature ($\sim 0.6$ kcal mol$^{-1}$), meaning that DTNN potential-energy surfaces can be utilized to calculate accurate molecular thermodynamic properties by virtue of Monte Carlo simulations.

Supplementary Figs 1 and 2 illustrate how the performance of DTNN depends on the number of employed reference calculations and refinement steps (Supplementary Discussion). The ability of DTNN to accurately describe equilibrium structures within the GDB-9 database and MD trajectories of selected molecules of chemical relevance demonstrates the feasibility of developing a universal machine learning architecture that can capture compositional as well as configurational degrees of freedom in the vast chemical space. While the employed architecture of the DTNN is universal, the learned coefficients are different for GDB-9 and MD trajectories of single molecules.

**Local chemical potential**. Beyond predicting accurate energies, the true power of DTNN lies in its ability to provide novel quantum-chemical insights. In the context of DTNN, we define a local chemical potential $\Omega_A^M(\mathbf{r})$ as an energy of a certain atom type $A$, located at a position $\mathbf{r}$ in the molecule $M$. While the DTNN models the interatomic interactions, we only allow the atoms of the molecule act on the probe atom, while the probe does not influence the molecule. The spatial and chemical sensitivity provided by our DTNN approach is shown in Fig. 1e for a variety of fundamental molecular building blocks. In this case, we employed hydrogen as a test charge, while the results for $\Omega_{C,N,O}^M(\mathbf{r})$ are shown in Fig. 2. Despite being trained only on total energies of molecules, the DTNN approach clearly grasps fundamental chemical concepts such as bond saturation and different degrees of aromaticity. For example, the DTNN model predicts the $C_6O_3H_6$ molecule to be 'more aromatic' than benzene

or toluene (Fig. 1e). Remarkably, it turns out that $C_6O_3H_6$ does have higher ring stability than both benzene and toluene and DTNN predicts it to be the molecule with the most stable aromatic carbon ring among all molecules in the GDB-9 database (Fig. 3). Further chemical effects learned by the DTNN model are shown in Fig. 2 that demonstrates the differences in the chemical potential distribution of H, C, N and O atoms in benzene, toluene, salicylic acid and malonaldehyde. For example, the chemical potentials of different atoms over an aromatic ring are qualitatively different for H, C, N and O atoms—an evident fact for a trained chemist. However, the subtle chemical differences described by DTNN are accompanied by chemically accurate predictions—a challenging task for humans.

Because DTNN provides atomic energies by construction, it allows us to classify molecules by the stability of different building blocks, for example aromatic rings or methyl groups. An example of such classification is shown in Fig. 3, where we plot the molecules with most stable and least stable carbon aromatic rings in GDB-9. The distribution of atomic energies is shown in Supplementary Fig. 3, while Supplementary Fig. 4 lists the full stability ranking. The DTNN classification leads to interesting stability trends, notwithstanding the intrinsic non-uniqueness of atomic energy partitioning. However, unlike atomic projections employed in electronic-structure calculations, the DTNN approach has a firm foundation in statistical learning theory. In quantum-chemical calculations, every molecule would correspond to a different partitioning depending on its self-consistent electron density. In contrast, the DTNN approach learns the partitioning on a large molecular dataset, generating a transferable and global 'dressed atom' representation of molecules in chemical space. Recalling that DTNN exhibits errors below 1 kcal mol$^{-1}$, the classification shown in Fig. 3 can provide useful guidance for the chemical discovery of molecules with desired properties. Analytical gradients of the DTNN model with respect to chemical composition or $\Omega_A^M(\mathbf{r})$ could also aid in the exploration of chemical compound space[37].

**Energy predictions for isomers**. The quantitative accuracy achieved by DTNN and its size extensivity paves the way to the calculation of configurational and conformational energy differences—a long-standing challenge for machine learning approaches[7,12,13,38]. The reliability of DTNN for isomer energy predictions is demonstrated by the energy distribution in Fig. 4 for molecular isomers with $C_7O_2H_{10}$ chemical formula (a total of 6,095 isomers in the GDB-9 data set).

Training a common model for chemical as well as conformational freedoms requires a more complex model. Furthermore, it comes with technical challenges like sampling and multiscale issues since the MD trajectories form clusters of small variation within the chemical compound space. As a proof of principle, we trained the DTNN to predict various MD trajectories of the $C_7O_2H_{10}$ isomers. To this end, we calculated short MD trajectories of 5,000 steps each for 113 randomly picked isomers as well as consistent total energies for all equilbrium structures. The training set is composed of all isomers in equilibrium as well as 50% of each MD trajectory. The remaining MD calculations are used for validation and testing. Despite the added complexity, our model achieves a mean absolute error of 1.7 kcal mol$^{-1}$.

**Discussion**
DTNNs provide an efficient way to represent chemical environments allowing for chemically accurate predictions. To this end, an implicit, atom-centered basis is learned from reference calculations. Employing this representation, atoms can be embedded in their chemical environment within a few refinement

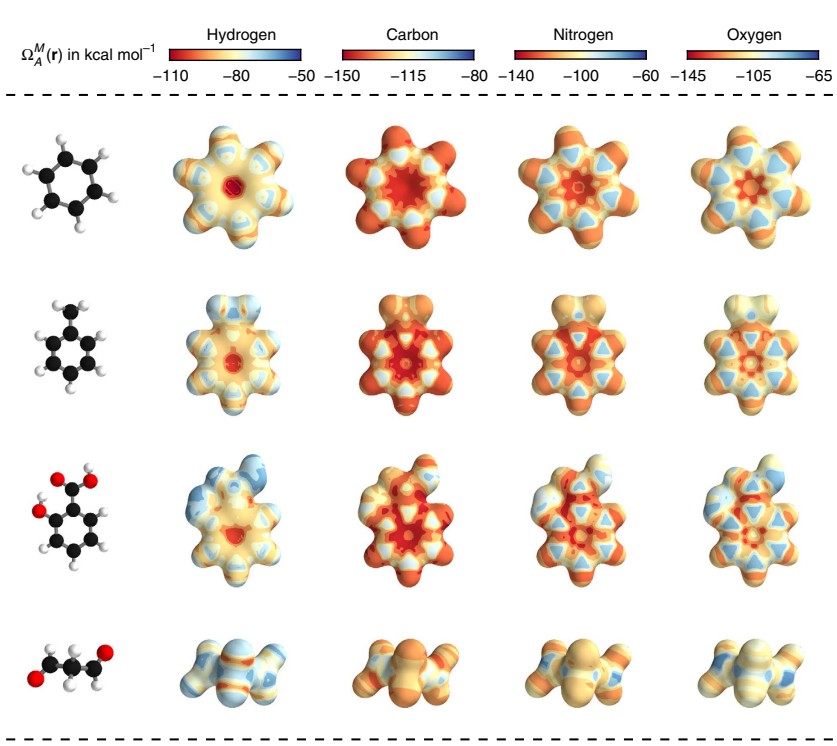

**Figure 2 | Chemical potentials $\Omega_A^M(\mathbf{r})$ for $A = \{C, N, O, H\}$ atoms.** The isosurface was generated for $\sum_i \|\mathbf{r} - \mathbf{r}_i\|^{-2} = 3.8\ \text{Å}^{-2}$ (the index $i$ is used to sum over all atoms of the corresponding molecule). The molecules shown are (in order from top to bottom of the figure): benzene, toluene, salicylic acid and malondehyde. Atom colouring: carbon = black, hydrogen = white, oxygen = red.

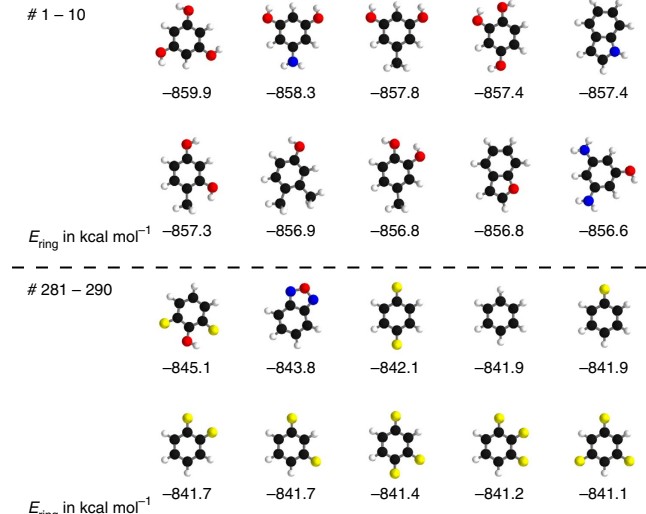

**Figure 3 | Classification of molecular carbon ring stability.** Shown are 20 molecules (10 most stable and 10 least stable) with respect to the energy of the carbon ring predicted by the DTNN model. Atom colouring: carbon = black; hydrogen = white; oxygen = red; nitrogen = blue; fluorine = yellow.

steps. Furthermore, DTNNs have the advantage that the embedding is built recursively from pairwise distances. Therefore, all necessary invariances (translation, rotation, permutation) are guaranteed to be exploited by the model. In addition, the learned embedding can be used to generate alchemical reaction paths (Supplementary Fig. 5).

In previous approaches, potential-energy surfaces were constructed by fitting many-body expansions with neural

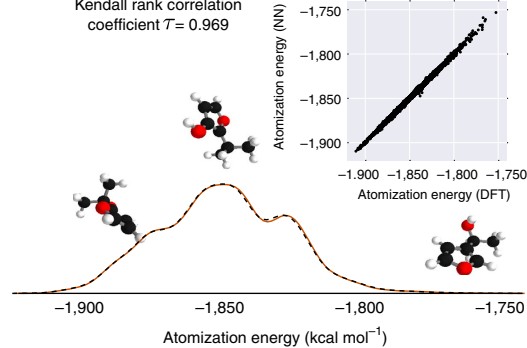

**Figure 4 | Isomer energies with chemical formula $C_7O_2H_{10}$.** DTNN trained on the GDB-9 database is able to acurately discriminate between 6,095 different isomers of $C_7O_2H_{10}$, which exhibit a non-trivial spectrum of relative energies.

networks[39–41]. However, these methods require a separate NN for each non-equivalent many-body term in the expansion. Since DTNN learns a common basis in which the atom interact, higher-order interactions can obtained more efficiently without separate treament.

Approaches like smooth overlap of atomic positions[16,17] or manually crafted atom-centered symmetry functions[18,19,42] are, like DTNN, based on representing chemical environments. All these approaches have in common that size-extensivity regarding the number of atoms is achieved by predicting atomic energy contributions using a non-linear regression method (for example, neural networks or kernel ridge regression). However, the previous approaches have a fixed set of basis functions describing the atomic environments. In contrast, DTNNs are able to adapt to the problem at hand in a

data-driven fashion. Beyond the obvious advantage of not having to manually select symmetry functions and carefully tune hyper-parameters of the representation, this property of the DTNN makes it possible to gain quantum-chemical insights by analysing the learned representation.

Obviously, more work is required to extend this predictive power for larger molecules, where the DTNN model will have to be combined with a reliable model for long-range interatomic (van der Waals) interactions. The intrinsic interpolation smoothness achieved by the DTNN model can also be used to identify molecules with peculiar electronic structure. Supplementary Fig. 6 shows a list of molecules with the largest DTNN errors compared with reference DFT calculations. It is noteworthy that most molecules in this figure are characterized by unconventional bonding and the electronic structure of these molecules has potential multi-reference character. The large prediction errors could stem from these molecules being not sufficiently represented by the training data. On the other hand, DTNN predictions might turn out to be closer to the correct answer because of its smooth interpolation in chemical space. Higher-level quantum-chemical calculations would be required to investigate this interesting hypothesis in the future.

We have proposed and developed a deep tensor neural network that enables understanding of quantum-chemical many-body systems beyond properties contained in the training dataset. The DTNN model is scalable with molecular size, efficient, and achieves uniform accuracy of 1 kcal mol$^{-1}$ throughout compositional and configuration space for molecules of intermediate size. The DTNN model leads to novel insights into chemical systems, a fact that we illustrated on the example of relative aromatic ring stability, local molecular chemical potentials, relative isomer energies and the identification of molecules with peculiar electronic structure.

Many avenues remain for improving the DTNN model on multiple fronts. Among these we mention the extension of the model to increasingly larger molecules, predicting atomic forces and frequencies, and non-extensive electronic and optical properties. We propose the DTNN model as a versatile framework for understanding complex quantum-mechanical systems based on high-throughput electronic structure calculations.

## Methods

**Reference data sets.** We employ two subsets of the GDB database[34], referred to in this paper as GDB-7 and GDB-9. GDB-7 contains 7,211 molecules with up to seven heavy atoms out of the elements C, N, O, S and Cl, saturated with hydrogen[12]. Similarly, GDB-9 includes 133,885 molecules with up to 9 heavy atoms out of C, O, N, F (ref. 36). Both data sets include calculations of atomization energies employing density-functional theory[43] with the PBE0 (ref. 44) and B3LYP (ref. 45–49) exchange-correlation potential, respectively.

The molecular dynamics trajectories are calculated at a temperature of 500 K and resolution of 0.5 fs using density-functional theory with the PBE exchange-correlation potential[50]. The data sets for benzene, toluene, malonaldehyde and salicylic acid consist of 627, 442, 993 and 320 k time steps, respectively. In the presented experiments, we predict the potential energy of the MD geometries.

**Details on the deep tensor neural network model.** The molecular energies of the various data sets are predicted using a deep tensor neural network. The core idea is to represent atoms in the molecule as vectors depending on their type and to subsequently refine the representation by embedding the atoms in their neighbourhood. This is done in a sequence of interaction passes, where the atom representations influence each other in a pair-wise fashion. While each of these refinements depends only on the pair-wise atomic distances, multiple passes enable the architecture to also take angular information into account. Because of this decomposition of atomic interactions, an efficient representation of embedded atoms is learned following quantum-chemical principles.

In the following, we describe the deep tensor neural network step-by-step, including hyper-parameters used in our experiments.

1. Assign initial atomic descriptors

We assign an initial coefficient vector to each atom $i$ of the molecule according to its nuclear charge $Z_i$:

$$\mathbf{c}_i^{(0)} = \mathbf{c}_{Z_i} \in R^B, \tag{4}$$

where $B$ is the number of basis functions. All presented models use atomic descriptors with 30 coefficients. We initialize each coefficient randomly following $\mathbf{c}_z \sim \mathcal{N}(0, 1/\sqrt{B})$.

2. Gaussian feature expansion of the inter-atomic distances

The inter-atomic distances $D_{ij}$ are spread across many dimensions by a uniform grid of Gaussians

$$\hat{\mathbf{d}}_{ij} = \left[ \exp\left( -\frac{(D_{ij} - (\mu_{\min} + k\Delta\mu))^2}{2\sigma^2} \right) \right]_{0 \le k \le \mu_{\max}/\Delta\mu}, \tag{5}$$

with $\Delta\mu$ being the gap between two Gaussians of width $\sigma$.

In our experiments, we set both to 0.2 Å. The centre of the first Gaussian $\mu_{\min}$ was set to $-1$, while $\mu_{\max}$ was chosen depending on the range of distances in the data (10 Å for GDB-7 and benzene, 15 Å for toluene, malonaldehyde and salicylic acid and 20 Å for GDB-9).

3. Perform $T$ interaction passes

Each coefficient vector $\mathbf{c}_i^{(t)}$, corresponding to atom $i$ after $t$ passes, is corrected by the interactions with the other atoms of the molecule:

$$\mathbf{c}_i^{(t+1)} = \mathbf{c}_i^{(t)} + \sum_{j \ne i} \mathbf{v}_{ij}. \tag{6}$$

Here, we model the interaction $v$ as follows:

$$\mathbf{v}_{ij} = \tanh\left[ W^{\text{fc}}\left( (W^{\text{cf}}\mathbf{c}_j + \mathbf{b}^{\text{f}_1}) \circ (W^{\text{df}}\hat{\mathbf{d}}_{ij} + \mathbf{b}^{\text{f}_2}) \right) \right], \tag{7}$$

where the circle ($\circ$) represents the element-wise matrix product. The factor representation in the presented models employs 60 neurons.

4. Predict energy contributions

Finally, we predict the energy contributions $E_i$ from each atom $i$. Employing two fully-connected layers, for each atom a scaled energy contribution $\hat{E}_i$ is predicted:

$$\mathbf{o}_i = \tanh\left( W^{\text{out}_1}\mathbf{c}_i^{(T)} + \mathbf{b}^{\text{out}_1} \right) \tag{8}$$

$$\hat{E}_i = W^{\text{out}_2}\mathbf{o}_i + \mathbf{b}^{\text{out}_2} \tag{9}$$

In our experiments, the hidden layer $\mathbf{o}_i$ possesses 15 neurons. To obtain the final contributions, $\hat{E}_i$ is shifted to the mean $E_\mu$ and scaled by the s.d. $E_\sigma$ of the energy per atom estimated on the training set.

$$E_i = E_\sigma \hat{E}_i + E_\mu \tag{10}$$

This procedure ensures a good starting point for the training.

5. Obtain the molecular energy $E = \sum_i E_i$

The bias parameters as well as $W^{\text{out}_2}$ are initially set to zero. All other weight matrices are initialized drawing from a uniform distribution according to (ref. 51). Neural network code is available.

The deep tensor neural networks have been trained for 3,000 epochs minimizing the squared error, using stochastic gradient descent with 0.9 momentum and a constant learning rate[52]. The final results are taken from the models with the best validation error in early stopping.

All DTNN models were trained and executed on an NVIDIA Tesla K40 GPU. The computational cost of the employed models depends on the number of reference calculations, the number of interaction passes as well as the number of atoms per molecule. The training times for all models and data sets are shown in Supplementary Table 2, ranging from 6 h for 5.768 reference calculations of GDB-7 with one interaction pass, to 162 h for 100,000 reference calculations of the GDB-9 data set with three interaction passes.

On the other hand, the prediction is instantaneous: all models predict examples from the employed data sets in <1 ms. Supplementary Fig. 7 shows the scaling of the prediction time with the number of atoms and interaction layers. Even for a molecule with 100 atoms, a DTNN with three interaction layers requires <5 ms for a prediction.

The prediction as well as the training steps scale linearly with the number of interaction passes and quadratically with the number of atoms, since the pairwise atomic distances are required for the interactions. For large molecules it is reasonable to introduce a distance cutoff. In that case, the DTNN will also scale linearly with the number of atoms.

**Computing and visualizing the local potentials of the DTNN.** Given a trained neural network as described in the previous section, one can extract the coefficients vectors $\mathbf{c}_i^{(t)}$ for each atom $i$ and each interaction pass $t$ for a molecule of interest. From each final representation $\mathbf{c}_i^{(T)}$, the energy contribution $E_i$ of the corresponding atom to the molecular energy can be obtained. Instead, we let the molecule act on a probe atom, described by its charge $z$ and the pairwise distances

$d_1,\ldots, d_n$ to the atoms of the molecule:

$$\mathbf{c}_{\text{probe}}^{(t+1)} = \mathbf{c}_{\text{probe}}^{(t)} + \sum_{j=1}^{n} \mathbf{v}_j, \tag{11}$$

with $\mathbf{v}_j = \tanh\big(W^{\text{fc}}((W^{\text{cf}}\mathbf{c}_j + \mathbf{b}^{\text{f}_1}) \circ (W^{\text{df}}\hat{\mathbf{d}}_j + \mathbf{b}^{\text{f}_2})\big)$. While this is equivalent to how the coefficient vectors of the molecule are corrected, here, the molecule does not get to be influenced by the probe. Now, the energy of the probe atom is predicted as usual from the final representation $\mathbf{c}_{\text{probe}}^{(T)}$.

Interpreting this as a local potential $\Omega_A^M(\mathbf{r})$ generated by the molecule, we can use the neural network to visualize the learned interactions as illustrated in Supplementary Fig. 8. The presented energy surfaces show the potential for different probe atoms plotted on an isosurface of $\sum_{i=1}^{n} d_i^{-2}$. We used Mayavi[53] for the visualization of the surfaces.

**Data availability.** The GDB-9 data set is available under the DOI 10.6084/m9.figshare.978904. All data sets used in this work are available at http://quantum-machine.org/datasets/.

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

## Acknowledgements

We thank Huziel Sauceda for providing molecular dynamics trajectories for $C_7O_2H_{10}$ isomers. K.T.S. and K.R.M. thank the Einstein Foundation for generously funding the ETERNAL project. Additional support was provided by the DFG (MU 987/20-1) and the Federal Ministry of Education and Research (BMBF) for the Berlin Big Data Center BBDC (01IS14013A). K.R.M. gratefully acknowledges the BK21 program funded by Korean National Research Foundation grant (No. 2012-005741). Part of this research was performed while the authors were visiting the Institute for Pure and Applied Mathematics (IPAM), which is supported by the National Science Foundation (NSF).

## Author contributions

K.T.S. conceived the DTNN, performed analyses and prepared the figures, K.T.S., F.A., K.R.M. and A.T. developed the theory, K.R.M. and A.T. designed the analyses, S.S. helped with the MD predictions, K.T.S., K.R.M. and A.T. wrote the paper. All authors discussed results and commented on the manuscript.

## Additional information

**Competing financial interests:** The authors declare no competing financial interests.

**Publisher's note**: 

