## [Peer Review File · Nature Communications]

Reviewers' comments:

Reviewer #1 (Remarks to the Author):

The work is a nice manifestation of the recent efforts of the computational chemistry community to create transferable, if possible universal, machine learning (ML) models, which can be used across compositional and configurational chemical space. The authors address several very important and difficult issues including size-extensivity and obtaining local, chemically relevant, information using ML.

The authors claim that their deep learning approach is able to solve above challenges with high accuracy and even ability to determine molecules with the peculiar electronic structure.

In the manuscript they convincingly shown that their DTNN are able to achieve size-extensivity in a nice and consistent manner by calculating and summing up atomic energy contributions. This potentially provides a straightforward way to analyze system at the atomic level. A semantic note: size-extensivity also implies that a method should properly scale with the number of electrons, which their method would not be able to do as it does not differentiate between charged and neutral molecules and all their models were trained on neutral molecules. The authors should also elaborate on several more points.

First, what advantages and disadvantages has their method in comparison with the SOAP (refs. 16 and 17), which should be size-extensive as well? Similarly, Behler and Parinello's HDNNs (Phys. Rev. Lett. 98, 146401 (2007) and refs. 18, 19 in the reviewed manuscript) calculate total energy by calculating atomic contributions, which can successfully be used for modeling organic reactions (Gastegger and Marquetand, J. Chem. Theory Comput. 11, 2187 (2015)). Similar comparisons should be provided for RS-HDMR-NNs (Manzhos and Carrington, J. Chem. Phys. 125, 084109 (2006)) and NNs for generalized PESs (Raff et al. J. Chem. Phys. 130, 184102 (2009)).

Second, the authors should clarify what exactly they mean by the 'total energy' for the molecule, which they also call the 'molecular energy'. I doubt that it is total energy from DFT calculations: most likely they used DFT atomization energy at 0 K without ZPVE corrections. In Fig. 1D the total energy spans from -40 to 0 kcal/mol, which can be neither DFT total energy nor atomization energy.

Third, the paper would benefit from additional discussion of scaling. How quickly do the errors rise for systems significantly larger than those in the training set? In Fig. S8 they shown timing for molecules with up to 100 atoms, but never discussed whether their approach breaks down for such systems in terms of accuracy. The later question is important to understand how size-extensive their method actually is. Related question is about timing: in aforementioned Fig. S8 the scaling does not look linear, but rather exponential. Obviously, it is still very fast, just ms for systems with up to 100 atoms, but it would be beneficial for the reader to see, for what system size calculations become noticeably slow (minutes?) to very expensive (months?). The main text could be improved by discussing benefits of ML in terms of computational time: milliseconds of prediction time with ML vs much slower QM calculations; probably a part of the Section 'Computational cost of training and prediction' in the Supplementary Materials can be used in the main text.

Fourth, if the authors want to demonstrate that DTNN can be used for creating a 'universal' ML model 'that can capture both compositional and configurational' differences, they have to create such a proof-of-principle ML model. They have to try to create a model, which is (almost) as accurate for both GDB-9 and MD trajectories as individual ML models are. Otherwise, their claim on page 3 that ability of the individual ML models to describe both GDB-9 or MD trajectories well 'demonstrates the feasibility of developing a universal' ML model contradicts with their following acknowledgment that 'while the employed architecture of the DTNN is universal, the learned coefficients are different for GDB-9 and MD trajectories of single molecules'. The universal model

should have not only the same structure, but also the same coefficients.

Fifth, section 'Quantum-Chemical Insights' is not very convincing. The 3D plots of the local chemical potentials (Fig. 1E, 2, S1) are nice pictures, but they provide little information. They lack any color scale, so it is impossible to say where the interaction is stronger or weaker. What is the physical nature of the interactions with the probe atom is also not clear: the authors said that they 'employed hydrogen as a test charge' and then did the same for C, N, O, but then in other places they talked about 'probe atom', not probe cation. The confusion can be understood as their DTNN do not contain any information about the probe charge let alone spin multiplicity. Thus, it makes impossible to verify DTNN local chemical potentials by quantum-chemical calculations. In addition, DTNN were not trained to represent correctly interaction of neutral molecules with ions or radicals. Example of a study, where ML provided real quantum mechanical observables of atoms is in von Lilienfeld et al. J. Phys. Chem. Lett. 6, 3309 (2015). Additionally, the authors discussed the stability of benzene rings in molecules by calculating their DTNN energies. Again, no verification with quantum-chemical methods of the stability ranking was performed. Such verification is necessarily and possible, e.g. by considering isodesmic reactions. Thus, the present manuscript appears to compare only molecular energies obtained with DTNN with those calculated with quantum-chemical methods. As for the molecules, which have the highest DTNN error, the authors correctly point out that it may be due to the failure of the particular DFT method to describe them. Nevertheless, the authors should also mention that the failure may as well be simply due to the underfitting, i.e. to due to the insufficient representation of similar molecules in the training set.

Other minor issues in the Supplementary Materials. Programs used to visualize molecules, isosurfaces should be cited. Citation for B3LYP has to include several more papers, at least Becke's paper from 1993. Sentence 'Due to this decomposition of atomic interactions, an efficient representation of embedded atoms is learned employing quantum chemical.' seems to be unfinished. Sentence 'where B is the number of basis functions, respectively neurons' does not make full sense. Equations has to be numbered. E_{μ} in the last equation on page 3 is never explained. Was RMSE or MAE minimized in the fitting? Fig. S2 and S3: change 'errorbars' to 'error bars'. Fig. S5: add units. Fig. S6, S7, S8: provide the size of the training set. Fig. S8: provide exact DTNN model used, i.e. was it trained on GDB-9? Section about the alchemical path and Fig. S7 were not really referred from the main text.

Finally, I feel that after above issues are resolved, the paper will provide a nice guide for researchers working in the field by setting standards for the next-generation size-extensive ML methods.

Reviewer #2 (Remarks to the Author):

The manuscript from Schuett et al. presents the use of deep tensor neural networks applied to quantum chemistry. In particular, the authors focus on learning molecular energies. They decompose molecular energies into atomic contributions, which allows them to look further at the contribution of fragments and spatially-resolved features. To the best of my knowledge, it is the first application of deep learning methodologies to predicting chemical properties. The methodology looks robust, the results of high quality, and the conclusions of interest.

The presentation of the paper is rather clear. Fig 1B illustrates well the main concept behind DTNNs. I do find the description of the nonlinear coupling at the end of the method section more difficult to read. Given that the paper will largely appeal to physicists and chemists who do not have background in deep learning, the introduction of the method might benefit from some further polishing.

- The message of the paper seems to be that using deep learning is superior to more conventional

ML techniques (kernel-based, standard neural networks, etc.) in its ability to provide more insight. I find some of the conclusions in the outlook somewhat debatable: (i) I don't see how the DTNN model provides new insight into the relative isomer energies; it merely predicts the DFT energies. (ii) Whether the molecules with peculiar electronic structure point at DFT or ML-prediction problems remains at this point speculative. The argument used by the authors against DFT could be also used against ML: exotic bonding and electronic structure might be seldom sampled in the training set and thus poorly interpolated.

- The application to an MD simulation is interesting, although the authors do not report on the transferability of the initial compositional ML model onto the MD simulations. Does that lead to poor results?

- Also related to MD: given the need of 10-100k reference conformations to get to accurate energies, it isn't clear at this stage the practicality of ML, since these reference conformations are likely enough to build the potential energy surface of the one molecule in vacuum. Using ML applied to compositional chemical space is useful because of the combinatorics of atoms, the conformational aspect here is not as convincing.

- It isn't so clear how to physically interpret what the iterations T mean here. Something like the size of the Coulomb matrix is much more straightforward in understanding how much information (correlations between neighboring atoms) goes into the ML. Can the authors provide a bit more insight?

- Why is there no MAE for 28 atoms in Fig 1C?

- Is there a typo in Fig 1B? "E1" is repeated several times (instead of "E2"...).

Reviewer #3 (Remarks to the Author):

The manuscript describes a new implementation of machine learning of molecular properties, by employing Deep Tensor Neural Networks. Using standard, off-the-shelf ML algorithms for atomistic systems is a challenge, because of the symmetries and the immense variability of the data. The authors present an innovative way of solving this problem, showing how to feed the input features (geometry and composition) into the currently very popular deep neural network scheme to extract atomic contributions. The chemical potential maps are also very interesting, providing exciting visualisation of the bonding. Accurate potential energy surfaces of small molecules are also very useful, although in this context DTNNs are just another example of such fitting methods, due to the fact that separate coefficients (and fitting) are needed for each molecule.

The paper makes an interesting read, it is scientifically sound, and it is a nice technological advance, but it somehow lacks the groundbreaking application. Achieving the "chemical accuracy" of 1 kcal/mol on the GDB7/GDB9 database is an important milestone and a fantastic demonstration of power of their method, albeit the database itself was not computed with at chemically accurate level. I would also like to point out that sub-1 kcal/mol MAE error on the GDB7 database has been reached before (see ref 33 for example).

In summary, I think the paper is high quality, and potentially will have a high impact on the field, but might be better placed in a less general journal.

Reviewer #1 (Remarks to the Author):

The work is a nice manifestation of the recent efforts of the computational chemistry community to create transferable, if possible universal, machine learning (ML) models, which can be used across compositional and configurational chemical space. The authors address several very important and difficult issues including size-extensivity and obtaining local, chemically relevant, information using ML.

The authors claim that their deep learning approach is able to solve above challenges with high accuracy and even ability to determine molecules with the peculiar electronic structure.

We thank the reviewer for a careful and detailed evaluation of our manuscript, which helped us to improve the presentation and better highlight the relevance of our DTNN approach to the chemistry and physics communities. In the revised manuscript, we have addressed all of the important comments raised by the reviewer. Below, we provide a concise reply to every comment.

In the manuscript they convincingly shown that their DTNN are able to achieve size-extensivity in a nice and consistent manner by calculating and summing up atomic energy contributions. This potentially provides a straightforward way to analyze system at the atomic level. A semantic note: size-extensivity also implies that a method should properly scale with the number of electrons, which their method would not be able to do as it does not differentiate between charged and neutral molecules and all their models were trained on neutral molecules. The authors should also elaborate on several more points.

We agree with the reviewer that the presented DTNN results concern compositional and configurational degrees of freedom for neutral molecules. Further extension of the methodology for charged molecules would be interesting, but is beyond the scope of the present paper.

First, what advantages and disadvantages has their method in comparison with the SOAP (refs. 16 and 17), which should be size-extensive as well? Similarly, Behler and Parinello's HDNNs (Phys. Rev. Lett. 98, 146401 (2007) and refs. 18, 19 in the reviewed manuscript) calculate total energy by calculating atomic contributions, which can successfully be used for modeling organic reactions (Gastegger and Marquetand, J. Chem. Theory Comput. 11, 2187 (2015)). Similar comparisons should be provided for RS-HDMR-NNs (Manzhos and Carrington, J. Chem. Phys. 125, 084109 (2006)) and NNs for generalized PESs (Raff et al. J. Chem. Phys. 130, 184102 (2009)).

We thank the reviewer for raising this important point. While we had acknowledged the important previous work on machine learning in our introduction, we agree with the reviewer that additional discussion is important to provide an optimal context for the readers of our paper. In the revised manuscript, we have added two paragraphs discussing the relation between our and previous work and included several additional references.

The developed DTNN approach is a substantial advance on multiple fronts:

- (1) DTNN learns an explicit atom-centered basis from reference ab initio calculations, preserving all molecular invariances by construction.*
- (2) DTNN enables accurate and efficient predictions for both compositional and configurational degrees of freedom.*
- (3) DTNN enables a wide range of novel insights into quantum-chemical systems.*

Each of these contributions is novel in itself, and their synergistic combination makes our manuscript particularly important for the broad readership of Nature Communications.

Second, the authors should clarify what exactly they mean by the

'total energy' for the molecule, which they also call the 'molecular energy'. I doubt that it is total energy from DFT calculations: most likely they used DFT atomization energy at 0 K without ZPVE corrections. In Fig. 1D the total energy spans from -40 to 0 kcal/mol, which can be neither DFT total energy nor atomization energy.

We apologize for this unclear presentation. It is the total energy of the DFT calculations. There was an offset of $-1.702e5$ kcal/mol missing in the plot axis. We fixed that.

Third, the paper would benefit from additional discussion of scaling. How quickly do the errors rise for systems significantly larger than those in the training set? In Fig. S8 they shown timing for molecules with up to 100 atoms, but never discussed whether their approach breaks down for such systems in terms of accuracy. The later question is important to understand how size-extensive their method actually is.

The DTNN output consists of atomic energies for each atom in their chemical environment. Such decomposition is fully consistent with the extensivity of molecular energies. Therefore, in principle the DTNN framework is able to scale by construction while staying with chemical environments where the underlying statistics is similar to the one represented in the training set.

The per-atom DTNN energy prediction and the fact that chemical interactions have a finite distance range means that the DTNN model will yield a constant error per atom upon increasing molecular size. To assess the effective range of chemical interactions we have imposed a distance cutoff to interatomic interactions of 3 \AA , yielding only a 0.1 kcal/mol increase in the error. However, this distance cutoff restricts only the direct interactions considered in the refinement steps. With multiple refinements, the effective cutoff increases by a factor of T due to indirect interactions over multiple atoms. Given large enough molecules, so that a reasonable distance cutoff can be chosen, scaling to larger molecules will require only to have well-represented local environments.

We have added this discussion in the revised manuscript.

Related question is about timing: in aforementioned Fig. S8 the scaling does not look linear, but rather exponential. Obviously, it is still very fast, just ms for systems with up to 100 atoms, but it would be beneficial for the reader to see, for what system size calculations become noticeably slow (minutes?) to very expensive (months?). The main text could be improved by discussing benefits of ML in terms of computational time: milliseconds of prediction time with ML vs much slower QM calculations; probably a part of the Section 'Computational cost of training and prediction' in the Supplementary Materials can be used in the main text.

We added information about the scaling to the supplement. DTNN scales quadratically with regard to the number of atoms, which can reduced to linear scaling when introducing a distance cutoff.

Fourth, if the authors want to demonstrate that DTNN can be used for creating a 'universal' ML model 'that can capture both compositional and configurational' differences, they have to create such a proof-of-principle ML model. They have to try to create a model, which is (almost) as accurate for both GDB-9 and MD trajectories as individual ML models are. Otherwise, their claim on page 3 that ability of the individual ML models to describe both GDB-9 or MD trajectories well 'demonstrates the feasibility of developing a universal' ML model contradicts with their following acknowledgment that 'while the employed architecture of the DTNN is universal, the learned coefficients are

different for GDB-9 and MD trajectories of single molecules'. The universal model should have not only the same structure, but also the same coefficients.

We clarified our statement in the manuscript. Not the trained model is universal, but the architecture of the network. Furthermore, we added an experiment with a model containing both chemical and conformational changes. Training a common model for chemical as well as conformational freedoms requires a more complex model. Furthermore, it comes with technical challenges such as sampling and multiscale issues since the MD trajectories form clusters of small variation within the chemical compound space. For these reasons, we expect the prediction accuracy to decrease.

Aiming towards such a universal model, we calculated short MD trajectories for 113 randomly picked isomers as well as consistent total energies for all equilibrium structures. We assign 50% of each MD trajectory as well as 50% of the isomers in equilibrium to the training set. Despite the added complexity, our model still achieves 1.7 kcal/mol when adding the MD trajectories to the data set. This test provides a proof-of-principle demonstration that DTNN is able to describe complex chemical spaces.

Fifth, section 'Quantum-Chemical Insights' is not very convincing. The 3D plots of the local chemical potentials (Fig. 1E, 2, S1) are nice pictures, but they provide little information. They lack any color scale, so it is impossible to say where the interaction is stronger or weaker. What is the physical nature of the interactions with the probe atom is also not clear: the authors said that they 'employed hydrogen as a test charge' and then did the same for C, N, O, but then in other places they talked about 'probe atom', not probe cation. The confusion can be understood as their DTNN do not contain any information about the probe charge let alone spin multiplicity. Thus, it makes impossible to verify DTNN local chemical potentials by quantum-chemical calculations. In addition, DTNN were not trained to represent correctly interaction of neutral molecules with ions or radicals. Example of a study, where ML provided real quantum mechanical observables of atoms is in von Lilienfeld et al. J. Phys. Chem. Lett. 6, 3309 (2015). Additionally, the authors discussed the stability of benzene rings in molecules by calculating their DTNN energies. Again, no verification with quantum-chemical methods of the stability ranking was performed. Such verification is necessarily and possible, e.g. by considering isodesmic reactions. Thus, the present manuscript appears to compare only molecular energies obtained with DTNN with those calculated with quantum-chemical methods.

Following the comments of the reviewer, we have restructured our manuscript slightly to clearly distinguish between the new insights versus comparisons between DTNN/QM energetics.

The local chemical potentials provide a way to visualize the distribution of interaction energies learned by the DTNN. The color scale is centered to the average energy on the surface and scaled to the maximum absolute deviation. Negative regions are colored red while positive regions are blue. The color scale is meaningful because the energies are on the scale of bonding energies. Obviously, our choice of coloring is not unique and others choices are possible. These aspects will be a topic of future work. We note in passing that many descriptors used in quantum chemistry or conceptual density-functional theory are also not quantum-mechanical observables, nevertheless these concepts have proven to be very influential in understanding chemistry.

The main insight from the color plots in Figs. 1 and 2 is to demonstrate the complexity of the learned DTNN potential and its resemblance to chemical concepts, such as aromaticity and bond saturation. Since DTNN utilizes only energies and has no chemical "biases" whatsoever, we consider it rather remarkable that we can use DTNN to extract chemically-relevant information. To our knowledge, our work is the first to achieve such non-trivial insights.

As for the molecules, which have the highest DTNN error, the authors

correctly point out that it may be due to the failure of the particular DFT method to describe them. Nevertheless, the authors should also mention that the failure may as well be simply due to the underfitting, i.e. to due to the insufficient representation of similar molecules in the training set.

This is correct. We added that possibility to the manuscript.

Other minor issues in the Supplementary Materials. Programs used to visualize molecules, isosurfaces should be cited. Citation for B3LYP has to include several more papers, at least Becke's paper from 1993. Sentence 'Due to this decomposition of atomic interactions, an efficient representation of embedded atoms is learned employing quantum chemical.' seems to be unfinished. Sentence 'where B is the number of basis functions, respectively neurons' does not make full sense. Equations has to be numbered. E_{μ} in the last equation on page 3 is never explained. Was RMSE or MAE minimized in the fitting? Fig. S2 and S3: change 'errorbars' to 'error bars'. Fig. S5: add units. Fig. S6, S7, S8: provide the size of the training set. Fig. S8: provide exact DTNN model used, i.e. was it trained on GDB-9? Section about the alchemical path and Fig. S7 were not really referred from the main text.

We fixed these issues in the supplement. Regarding Fig. S8: The model used for the speed test was untrained, since the timing does not depend on the parameter values but only on the number of atoms and interaction passes.

Finally, I feel that after above issues are resolved, the paper will provide a nice guide for researchers working in the field by setting standards for the next-generation size-extensive ML methods.

*We thank the reviewer for helping us to improve the presentation of our paper and fully agree that our manuscript sets “**standards for the next-generation size-extensive ML methods**”.*

Reviewer #2 (Remarks to the Author):

The manuscript from Schuett et al. presents the use of deep tensor neural networks applied to quantum chemistry. In particular, the authors focus on learning molecular energies. They decompose molecular energies into atomic contributions, which allows them to look further at the contribution of fragments and spatially-resolved features. To the best of my knowledge, it is the first application of deep learning methodologies to predicting chemical properties. The methodology looks robust, the results of high quality, and the conclusions of interest.

The presentation of the paper is rather clear. Fig 1B illustrates well the main concept behind DTNNs. I do find the description of the nonlinear coupling at the end of the method section more difficult to read. Given that the paper will largely appeal to physicists and chemists who do not have background in deep learning, the introduction of the method might benefit from some further polishing.

We thank the reviewer for a careful and detailed evaluation of our manuscript, which helped us to improve the presentation and better highlight the relevance of our DTNN approach to the chemistry and physics communities. In the revised manuscript, we have addressed all of the important comments raised by the reviewer. Below, we provide a concise reply to every comment.

- The message of the paper seems to be that using deep learning is superior to more conventional ML techniques (kernel-based, standard neural networks, etc.) in its ability to provide more insight. I find some of the conclusions in the outlook somewhat debatable: (i) I don't see how the DTNN model provides new insight into the relative isomer energies; it merely predicts the DFT energies.

Following the comments of the reviewer, we have restructured our manuscript slightly to clearly distinguish between the new insights versus comparisons between DTNN/QM energetics.

(ii) Whether the molecules with peculiar electronic structure point at DFT or ML-prediction problems remains at this point speculative. The argument used by the authors against DFT could be also used against ML: exotic bonding and electronic structure might be seldom sampled in the training set and thus poorly interpolated.

This is true. We have added this possibility to the paper.

- The application to an MD simulation is interesting, although the authors do not report on the transferability of the initial compositional ML model onto the MD simulations. Does that lead to poor results?

We clarified our statements in the manuscript. Not the trained model is universal, but the architecture of the network. Furthermore, we added an experiment with a model containing both chemical and conformational changes. Training a common model for chemical as well as conformational freedoms requires a more complex model. Furthermore, it comes with technical challenges such as sampling and multiscale issues since the MD trajectories form clusters of small variation within the chemical compound space. For these reasons, we expect the prediction accuracy to decrease.

Aiming towards a universal model, we calculated short MD trajectories for 113 randomly picked isomers as well as consistent total energies for all equilibrium structures. We assign 50% of each MD trajectory as well as 50% of the isomers in equilibrium to the training set. Despite the added complexity, our model still achieves 1.7 kcal/mol when adding the MD trajectories to the data set. This test provides a proof-of-principle demonstration that DTNN is able to describe complex chemical spaces.

- Also related to MD: given the need of 10-100k reference conformations to get to accurate energies, it isn't clear at this stage the practicality of ML, since these reference conformations are likely enough to build the potential energy surface of the one molecule in vacuum. Using ML applied to compositional chemical space is useful because of the combinatorics of atoms, the conformational aspect here is not as convincing.

The point here was to demonstrate the possibility to encode fine-grained configurational changes with the DTNN. The conformational aspect is also important for a possible extension to training on atomic forces which should reduce the required number of reference calculations.

- It isn't so clear how to physically interpret what the iterations T mean here. Something like the size of the Coulomb matrix is much more straightforward in understanding how much information (correlations between neighboring atoms) goes into the ML. Can the authors provide a bit more insight?

We added further explanation to the paper how the interaction passes encode increasingly complex interactions into the atom descriptors.

- Why is there no MAE for 28 atoms in Fig 1C?

There are no molecules with 28 atoms in the GDB-9 data set.

- Is there a typo in Fig 1B? "E1" is repeated several times (instead of "E2" ...).

Yes, that was a typo. We corrected that.

Reviewer #3 (Remarks to the Author):

The manuscript describes a new implementation of machine learning of molecular properties, by employing Deep Tensor Neural Networks. Using standard, off-the-shelf ML algorithms for atomistic systems is a challenge, because of the symmetries and the immense variability of the data. The authors present an innovative way of solving this problem, showing how to feed the input features (geometry and composition) into the currently very popular deep neural network scheme to extract atomic contributions. The chemical potential maps are also very interesting, providing exciting visualisation of the bonding. Accurate potential energy surfaces of small molecules are also very useful, although in this context DTNNs are just another example of such fitting methods, due to the fact that separate coefficients (and fitting) are needed for each molecule.

The paper makes an interesting read, it is scientifically sound, and it is a nice technological advance, but it somehow lacks the groundbreaking application. Achieving the "chemical accuracy" of 1 kcal/mol on the GDB7/GDB9 database is an important milestone and a fantastic demonstration of power of their method, albeit the database itself was not computed with at chemically accurate level. I would also like to point out that sub-1 kcal/mol MAE error on the GDB7 database has been reached before (see ref 33 for example).

In summary, I think the paper is high quality, and potentially will have a high impact on the field, but might be better placed in a less general journal.

We thank the reviewer for a careful evaluation of our manuscript and his/her praising of our results as "new", "innovative", "accurate", "interesting", "exciting", and "high quality". We certainly agree with this appreciation. We also note that our manuscript contains not just one application of the DTNN method, but at least four new applications that demonstrate the accuracy, efficiency, and novel insights into quantum-chemical systems. Motivated by the comments of the reviewers, we have also improved the context of our manuscript by discussing our advances in light of previous approaches (see the new Discussion section). This makes it easier for the reader to assess the novelty of our development and applications of the DTNN approach.

The "chemical accuracy" of 1 kcal/mol achieved by the DTNN model is not our main focus. This remarkable accuracy makes the DTNN model worthy of analysis and offers novel insights into quantum-chemical systems, which is one of the main points of the manuscript. To reiterate, the developed DTNN approach is a substantial advance on multiple fronts:

- (1) DTNN learns an explicit atom-centered basis from reference ab initio calculations, preserving all molecular invariances by construction.*
- (2) DTNN enables accurate and efficient predictions for both compositional and configurational degrees of freedom.*
- (3) DTNN enables a wide range of novel insights into quantum-chemical systems.*

Each of these contributions is novel in itself, and their synergistic combination makes our manuscript particularly important for the broad readership of Nature Communications.

Based on the positive comments of the three reviewers, we hope that our manuscript will be deemed suitable for publication in Nature Communications.

REVIEWERS' COMMENTS:

Reviewer #1 (Remarks to the Author):

The revised version is an improvement over the original version and authors' replies address my comments point-by-point. Nevertheless, couple of authors' modifications are still necessary to improve.

First, it is nice that the authors provide a description of the color scale for chemical potentials (caption to Fig. 2) requested in my previous comment. However, the authors should provide actual numbers with the units for the blue and red colors and specified what color scheme (rainbow?) they used with indicated color of the origin. Usually it is done by providing a stripe with the range of used colors and corresponding numbers (something like this: http://thumbnails-visually.netdna-ssl.com/high-resolution-global-topographic-map-of-moon_50290dafa5273_w1500.jpeg).

Second, text in the manuscript slightly differs from their reply to my comment: "We assign 50% of each MD trajectory as well as 50% of the isomers in equilibrium to the training set" in the reply, while "The training set is composed of the isomers in equilibrium as well as 50% of each MD trajectory" in the manuscript. From the text in the manuscript it is not clear whether they have included 100% or 50% of the equilibrium structures to the training set. Additionally, they show that their common network for compositional and configurational degrees of freedoms "achieves a mean absolute error of 1.7 kcal/mol", but I expect that this error is dominated by more numerous non-equilibrium than equilibrium structures, so it would be nice if the authors provided separately the mean absolute error for the equilibrium structures as well, if they have 50% structures not included to the training set.

Above modifications are minor and I believe can be done swiftly.

Reviewer #2 (Remarks to the Author):

The authors have addressed the comments I raised.

REVIEWERS' COMMENTS:

Reviewer #1 (Remarks to the Author):

The revised version is an improvement over the original version and authors' replies address my comments point-by-point. Nevertheless, couple of authors' modifications are still necessary to improve.

First, it is nice that the authors provide a description of the color scale for chemical potentials (caption to Fig. 2) requested in my previous comment. However, the authors should provide actual numbers with the units for the blue and red colors and specified what color scheme (rainbow?) they used with indicated color of the origin. Usually it is done by providing a stripe with the range of used colors and corresponding numbers (something like this: <http://thumbnails-visually.net> [16]).

In response to your comment, we put molecules being probed with the same atom on a common scale and added color maps with corresponding energies to each column in Fig. 2.

Second, text in the manuscript slightly differs from their reply to my comment: “We assign 50% of each MD trajectory as well as 50% of the isomers in equilibrium to the training set” in the reply, while “The training set is composed of the isomers in equilibrium as well as 50% of each MD trajectory” in the manuscript. From the text in the manuscript it is not clear whether they have included 100% or 50% of the equilibrium structures to the training set. Additionally, they show that their common network for compositional and configurational degrees of freedoms “achieves a mean absolute error of 1.7 kcal/mol”, but I expect that this error is dominated by more numerous non-equilibrium than equilibrium structures, so it would be nice if the authors provided separately the mean absolute error for the equilibrium structures as well, if they have 50% structures not included to the training set.

We apologize for the confusion. The text in the paper is correct: We put 100% of the equilibrium structures and 50% of each MD trajectory in the training set.